# Field-Induced Transversely Isotropic Shear Response of Ellipsoidal Magnetoactive Elastomers

**DOI:** 10.3390/ma14143958

**Published:** 2021-07-15

**Authors:** Sanket Chougale, Dirk Romeis, Marina Saphiannikova

**Affiliations:** Leibniz Institute of Polymer Research Dresden e. V., Hohe Strasse 6, 01069 Dresden, Germany; romeis@ipfdd.de (D.R.); grenzer@ipfdd.de (M.S.)

**Keywords:** magnetoactive elastomers, shear deformations, magneto-rheological effect, magnetic torque

## Abstract

Magnetoactive elastomers (MAEs) claim a vital place in the class of field-controllable materials due to their tunable stiffness and the ability to change their macroscopic shape in the presence of an external magnetic field. In the present work, three principal geometries of shear deformation were investigated with respect to the applied magnetic field. The physical model that considers dipole-dipole interactions between magnetized particles was used to study the stress-strain behavior of ellipsoidal MAEs. The magneto-rheological effect for different shapes of the MAE sample ranging from disc-like (highly oblate) to rod-like (highly prolate) samples was investigated along and transverse to the field direction. The rotation of the MAE during the shear deformation leads to a non-symmetric Cauchy stress tensor due to a field-induced magnetic torque. We show that the external magnetic field induces a mechanical anisotropy along the field direction by determining the distinct magneto-mechanical behavior of MAEs with respect to the orientation of the magnetic field to shear deformation.

## 1. Introduction

Magnetoactive elastomers (MAEs) are field-controllable materials whose mechanical properties can be manipulated by applying an external magnetic field. They comprise micron-sized magnetically soft/hard particles integrated into an elastomer matrix. MAEs are used in several engineering applications because of their exceptional macroscopic shape response to an applied magnetic field and a magneto-mechanical coupled behavior [1,2,3,4,5,6,7]. General applications of MAEs include actuators, sensors, adaptively tuned vibration absorbers, dampers, microfluid transport systems, adaptive engine mounts [8,9,10,11]. Automotive suspension bushing is a pioneer application of the MAE developed by Ginder et al. [8]. The adaptively tuned vibration absorbers utilize MAEs as variable-spring rate elements. The microfluid transport systems use the ability of MAEs to change their shape in an external magnetic field, and such changes drive the fluid through artificial vessels [5]. In addition to these applications, MAEs are also used in the biomedical applications such as magnetic fixator of eye retina, artificial cilia, active porous scaffold controlled by a magnetic field, adjustable active surface morphology etc. [4,12,13,14]. MAEs can also alter the material parameters such as elastic and shear moduli in the magnetic field. For instance, the field-induced increase and decrease in the elastic modulus of MAEs with respect to the orientation of the applied magnetic field are recently predicted [15]. The wide variety of applications of MAEs makes it crucially important to understand mechanics triggering the change in mechanical properties.

The MAEs can be differentiated based on the method of synthesis as isotropic and anisotropic MAEs. The application of a uniform external magnetic field during the cross-linking procedure leads to “chain-like” structures of magnetic particles inside the matrix, whereas application of the rotating magnetic field results in “plane-like” structures of magnetic particles [16,17,18,19]. These structures produce a mechanical anisotropy in the MAE and the magneto-mechanical behavior varies strongly with the spatial distribution of the magnetic particles. On the other hand, mechanically isotropic MAEs with random distribution of magnetic particles demonstrate a “transverse isotropy” in the presence of an external magnetic field [15]. Different theoretical approaches have been proposed to investigate the magneto-mechanical behavior of MAEs [20,21,22,23,24,25,26,27,28,29,30,31]. These approaches can be broadly divided into particle-interaction models, micro-scale and macro-scale continuum models [32]. The micro-scale continuum models fully resolve the local magnetic and mechanical field with the help of a continuum formulation of a coupled magneto-mechanical boundary value problem [26,33]. The particle-interaction models do not resolve the local fields instead ulitizing effective pair-wise interactions. The macro-scale models predict the material behaviour of MAEs without resolving the microstructure [34,35,36]. The micro-scale and particle-interaction models describe the effect of local particle arrangement on the magneto-mechanical behaviour of MAEs, while macro-scale continuum models address the issue of the sample shape of MAEs. It is well known from theoretical [15,37] and experimental studies [38,39] that the shape of MAEs plays a crucial role in the magneto-mechanical behavior. Thus, models of MAEs that account for effects from microstructure alone or only from its macroscopic shape cannot fully describe the effective material behaviour. Accordingly, the complex interplay between the microstructure and macroscopic shape creates the necessity of a unified approach bridging different scales. For example, the cascading mean-field description of the magnetization field in MAE composites is recently proposed by splitting the field into three contributions on different scales [40]. As the magnetization field provides a detailed description of local and global effects, this strategy allows to decouple the short-range contributions from the long-range contributions to the magnetization field in the MAE.

As a path towards the effective material model of the MAE, one needs a theoretical framework that allows considering the magneto-mechanical effects emerging from the microstructure and shape of MAEs. Such theoretical framework is provided by a unified continuum-mechanics and microscopic approach [22], which we apply in this work to study the magneto-mechanical response of MAEs to shear deformations. This unification of macroscopic and microscopic approaches can lead towards a generalized “analytical” material model of MAEs. In our previous study, the particle-interaction model and micro-scale continuum model have been compared [28]. The comparison shows a very good agreement between both modeling strategies, especially for isotropic particle distribution. Thus, the analytical model based on dipole-dipole interactions presented in this work significantly reduces the computation resources otherwise required in the micro-scale continuum modeling. The unified approach is based originally on the “ellipsoidal” approximation to the sample shape, though some limitations of this approximation have been recently reported [24]. Recent works provide the extension of unified approach to any arbitrary sample shapes [40,41].

The present work focuses mainly on investigating the material behavior of MAEs during the shear deformation and predicts the magneto-rheological effect, defined as the change in the shear moduli in the presence of an external magnetic field. The shear deformation along the field direction, perpendicular to the field direction, and in the isotropic plane (plane perpendicular to the field direction) is studied. Following the different geometries of shear deformations, we examine the effect of field-induced magnetic torque and its consequences on the symmetry of the Cauchy stress tensor. In the initial state before applying the magnetic field or any deformation, the MAE has a shape of an ellipsoid of revolution with two equal semi-axes B=C, as shown in Figure 1. We consider a random distribution of spherical micron-sized magnetically soft particles in the MAE.

## 2. Material Model

We define $\vec{X}$ as the position vector of a material point in the reference configuration (undeformed) and $\vec{x}$ as the position vector in the current configuration (deformed), then the deformation gradient tensor is defined as $F_{mech}=\frac{\partial \vec{x}}{\partial \vec{X}}$ [42]. The corresponding right and left-Cauchy deformation tensors are $C_{mech}=F_{mech}^{T}\cdot F_{mech}$, $b_{mech}=F_{mech}\cdot F_{mech}^{T}$ [42,43,44]. The principal invariants of the left Cauchy deformation tensor are [44]
(1)$$I_{1}=\mathrm{tr}\left(b_{mech}\right),I_{2}=\frac{1}{2}\left(\mathrm{tr}{\left(b_{mech}\right)}^{2}-\mathrm{tr}\left(b_{mech}^{2}\right)\right),I_{3}=\det\left(b_{mech}\right)$$

As rubber-like materials are incompressible, i.e., $I_{3}=1$, and conventionally represented by the Neo-Hookean model, we assume MAEs to be modeled by a Neo-Hookean solid in the absence of a magnetic field. Thus, the elastic free energy per unit volume of the MAE is given as
(2)$$\psi_{el}=\frac{G}{2}\left(I_{1}-3\right)$$
where G is the shear modulus in Pa. The assumption of linear magnetization provides that the magnetization of particles $\vec{M}$ is proportional to the total magnetic field $\vec{H}$
(3)$$\vec{M}=\chi \vec{H}$$
where χ is the magnetic susceptibility. The total magnetic field $\vec{H}$ has been derived in [41]
(4)$$\vec{H}=\vec{H_{0}}+\left(G-n_{p}I\right)\cdot \vec{M}$$
where $\vec{H_{0}}$ is the applied magnetic field, $n_{p}=\frac{1}{3}$ is the isotropic demagnetizing factor of a spherical magnetically soft particle, and **I** is the identity tensor. The tensor $G=G_{macro}+G_{micro}$ is defined as a sum of macroscopic contributions from the long-range interactions between magnetized particles and microscopic contributions that take into account the local particle distribution [41]. In this work, we consider the isotropic distribution of magnetic particles inside an elastomer matrix, for which the contributions from local particle arrangement are vanishing, $G_{micro}=0$ [41,45,46,47]. With this, we rewrite Equation (Equation 4)
(5)$$\vec{H}=\vec{H_{0}}+\left(G_{macro}-n_{p}I\right)\cdot \vec{M}$$

From Equations (Equation 3) and (Equation 5) we derive an expression for the magnetization $\vec{M}$ as a function of the applied magnetic field $\vec{H_{0}}$
(6)$$\vec{M}={\left(I-\chi \left(G_{macro}-n_{p}I\right)\right)}^{-1}\cdot \chi \vec{H_{0}}$$

Here, $G_{macro}=\phi \left(n_{p}I-J\right)$, ϕ is the volume fraction of magnetic particles, **J** is the shape-dependent demagnetizing tensor. The Equation (Equation 6) can be further simplified by substituting the expression for $G_{macro}$ as
(7)$$\vec{M}={\left(RI+\phi J\right)}^{-1}\cdot \vec{H_{0}}$$
where $R=\chi^{-1}+n_{p}-\phi n_{p}$. Further, the magnetic energy due to the interactions between the magnetized particles in the case of linear magnetization behaviour reads [22]
(8)$$\psi_{mag}=-\frac{\mu_{0}\phi }{2}\left(\vec{M}\cdot \vec{H_{0}}\right)$$
where $\mu_{0}$ = $4\pi \times 10^{-7}$ NA${}^{-2}$ is the permeability of vacuum. The magnetization $\vec{M}$ depends on two aspect ratios $\gamma_{1}=\sqrt{\frac{e_{1}}{e_{2}}}$ and $\gamma_{2}=\sqrt{\frac{e_{1}}{e_{3}}}$ of an ellipsoidal MAE, where $e_{1},e_{2},e_{3}$ are the squares of principal stretches along the x,y and *z*-axis, respectively. Substituting the expression for the magnetization $\vec{M}$ from Equation (Equation 7) into the magnetic energy in Equation (Equation 8), we receive the magnetic energy of an ellipsoidal MAE as
(9)$$\psi_{mag}=-\frac{\mu_{0}\phi }{2}\left(\vec{{\left(H_{0}\right)}_{e}}\cdot J^{*}\cdot \vec{{\left(H_{0}\right)}_{e}}\right)$$
where $J^{*}=(RI+\phi J_{e}{)}^{-1}$, $J_{e}$ is the demagnetizing tensor of an ellipsoid. By adapting a coordinate system which coincides with the principal axes of an ellipsoid, the demagnetizing tensor $J_{e}$ can be represented as
(10)$$J_{e}=\left[\begin{array}{ccc} J_{a} & 0 & 0\\ 0 & J_{b} & 0\\ 0 & 0 & J_{c} \end{array}\right]$$
where $J_{a}$, $J_{b}$, $J_{c}$ are the demagnetizing factors of a general ellipsoid and $J_{a}+J_{b}+J_{c}=1$ [48]. Similarly, the applied magnetic field vector $\vec{H_{0}}$ can be represented in the principal axes of an ellipsoidal MAE as
(11)$$\vec{{\left(H_{0}\right)}_{e}}=Q^{T}\vec{H_{0}}$$
where *Q* is the rotation matrix that rotates between the main coordinate system and principal axes of an ellipsoid
(12)$$Q=\left[\begin{array}{ccc} \cos\alpha \cos\beta & \cos\alpha \sin\beta \sin\delta -\sin\alpha \cos\delta & \cos\alpha \sin\beta \cos\delta +\sin\alpha \sin\delta \\ \sin\alpha \cos\beta & \sin\alpha \sin\beta \sin\delta +\cos\alpha \cos\delta & \sin\alpha \sin\beta \cos\delta -\cos\alpha \sin\delta \\ -\sin\beta & \cos\beta \sin\delta & \cos\beta \cos\delta \end{array}\right]$$

Here α,β and δ are angles between the principal axes $e_{x},e_{y},e_{z}$ of an ellipsoidal MAE and principal directions x,y,z in the cartesian coordinate system, respectively. Combining Equations (Equation 2) and (Equation 9), the total free energy per unit volume of an ellipsoidal MAE becomes
(13)$$\psi_{MAE}=\frac{G}{2}\left(I_{1}-3\right)-\frac{\mu_{0}\phi }{2}\left(\vec{{\left(H_{0}\right)}_{e}}\cdot J^{*}\cdot \vec{{\left(H_{0}\right)}_{e}}\right)$$

The corresponding Cauchy stress tensor can be derived from the free energy of an ellipsoidal MAE as [15,42]
(14)$$\sigma =-pI+Gb_{mech}\;-\frac{\mu_{0}\phi }{2}\left(\vec{{\left(H_{0}\right)}_{e}}⊗\vec{{\left(H_{0}\right)}_{e}}:\frac{\partial J^{*}}{\partial F_{mech}}+J^{*}:\frac{\partial \left(\vec{{\left(H_{0}\right)}_{e}}⊗\vec{{\left(H_{0}\right)}_{e}}\right)}{\partial F_{mech}}\right)\cdot F_{mech}^{T}$$
where *p* is the hydrostatic pressure. The contribution of derivatives of $\vec{{\left(H_{0}\right)}_{e}}⊗\vec{{\left(H_{0}\right)}_{e}}$ with respect to $F_{mech}$ in the Cauchy stress tensor is non-symmetric. The symmetry of the Cauchy stress tensor is the result of applying the conservation of angular momentum to an infinitesimal material element. But in this work, we consider a macroscopic MAE sample coupled to an external magnetic field. The deformation gradient tensor **F** can be decomposed as F=R·U where **R** is called the rotation tensor and **U** is the right stretch tensor. In the case of R≠I, which is generally true for the shear deformation, MAEs rotate in the main axis system. It is known in the literature [49] that an ellipsoidal magnetic body (in our case, an ellipsoidal MAE) in the presence of a uniform external magnetic field experiences a magnetic torque due to the angle between magnetization and applied magnetic field. The presence of an external magnetic field yields additional magnetic couple $\mu_{0}\phi \left(\vec{M}\times \vec{H_{0}}\right)$ [32]. But the derivation of the Cauchy stress tensor assumes that there are no body moments. Thus, the magnetic torque acting on an ellipsoidal MAE affects the symmetry of the Cauchy stress tensor. The balance of angular momentum requires that $\sigma +\mu_{0}\phi \left(\vec{M}⊗\vec{H_{0}}\right)$ is symmetric or equivalently $\hat{\epsilon }:\sigma +\mu_{0}\phi \left(\vec{M}\times \vec{H_{0}}\right)=\vec{0}$, where $\hat{\epsilon }$ is the third order permutation tensor [35,50]. We introduce an antisymmetric tensor
(15)$$\tau =\mu_{0}\phi \left({\left(\vec{M}⊗\vec{H_{0}}\right)}^{T}-\vec{M}⊗\vec{H_{0}}\right)=\left[\begin{array}{ccc} 0 & -\tau_{xy} & -\tau_{xz}\\ \tau_{xy} & 0 & -\tau_{yz}\\ \tau_{xz} & \tau_{yz} & 0 \end{array}\right]$$

Here, $\tau_{ij}=\mu_{0}\phi \left(M_{i}{\left(H_{0}\right)}_{j}-M_{j}{\left(H_{0}\right)}_{i}\right),i\neq j$ is the magnitude of the magnetic torque. From the above symmetry conditions, we can write
(16)$$\sigma -\sigma^{T}=\tau$$

The Equations (Equation 15) and (Equation 16) yield the following relation
(17)$$\sigma_{ij}-\sigma_{ji}=-\tau_{ij}$$

For i≠j, the difference between the shear stress components $\sigma_{ji}-\sigma_{ij}$ is exactly equal to the magnitude of the magnetic torque $|\vec{\tau }|$ exerted by an external magnetic field. A detailed discussion of the magnetic torque acting on an MAE sample is presented in the next section. From the relations presented in Equation (Equation 17), it follows that $\sigma_{ij}+\frac{1}{2}\tau_{ij}=\sigma_{ji}-\frac{1}{2}\tau_{ij}$. Thus, the total symmetric stress tensor **T** can be constructed for a general case as
(18)$$T=\sigma -\frac{1}{2}\tau$$

One can also introduce the total symmetric stress tensor as $T=\frac{1}{2}\left(\sigma +\sigma^{T}\right)$ from Equations (Equation 16) and (Equation 18).

## 3. Magnetic Torque

Under the influence of an applied uniform magnetic field, an ellipsoidal MAE suspended freely in the field deforms and rotates to align its longest axis with the field direction (for details see Appendix A). It reflects that the MAE experiences a field-induced macroscopic magnetic torque in the presence of an applied magnetic field [51]. The torque exerted by the magnetic field on an MAE sample can be obtained with the help of magnetic energy in the equilibrium. The torque exerted by the magnetic field per unit volume is defined as [49,52,53]
(19)$$\vec{\tau }=\mu_{0}\phi \left(\vec{M}\times \vec{H_{0}}\right)$$

Consider a prolate MAE sample suspended freely in an external magnetic field, as shown in Figure 2a. In this case, the magnetic torque $\vec{\tau }$ is directed towards the *z*-axis and rotates a prolate MAE sample in an anticlockwise direction to align the symmetry axis along the field direction *x*. The magnetic energy per unit volume of an ellipsoidal MAE tilted under the angle α to the applied magnetic field in the x−y plane is obtained from Equation (Equation 9)
(20)$$\psi_{mag}=-\frac{\mu_{0}\phi H_{0}^{2}}{2}\left(\frac{\cos^{2}\alpha }{R+\phi J_{a}}+\frac{\sin^{2}\alpha }{R+\phi J_{b}}\right)$$
The magnitude of the torque $\vec{\tau }$ per unit volume can be determined by taking the derivative of the magnetic energy over the rotation angle α [51]
(21)$$\tau_{xy}=-\frac{\partial \psi_{mag}}{\partial \alpha }=2\zeta \sin\alpha \cos\alpha \left(\frac{J_{a}-J_{b}}{\left(R+\phi J_{a}\right)\left(R+\phi J_{b}\right)}\right)$$
where $\zeta =\left(\mu_{0}\phi^{2}{H_{0}}^{2}\right)/2$. The total magnetic field in Equation (Equation 4) is influenced by the competing tendencies of the magnetization $\vec{M}$ to orient parallel to the applied magnetic field and to orient along to the longest axis of an MAE sample because it has the lowest demagnetizing factor [49]. Since the longest axis of an ellipsoid has the smallest demagnetizing factor and vice versa, the difference $J_{a}-J_{b}$ is negative for prolate MAEs. It ensures that the torque $\vec{\tau }$ is restoring and it tries to align the symmetry axis of a prolate MAE along the field direction. On the other hand, the difference $J_{a}-J_{b}$ is positive for oblate MAEs. It means that the torque is not restoring in this case and acts so that the symmetry axis of an oblate MAE attempts to align perpendicular to the field direction. Figure 2b shows the dimensionless magnitude of the torque $\tau_{m}$ as a function of the rotation angle α for different aspect ratios $\gamma_{0}$. The magnitude of the torque is maximal at $\alpha =45^{\circ }$ and zero at $\alpha =90^{\circ }$ irrespective of the aspect ratio. It appears to be an equilibrium state at $\alpha =90^{\circ }$ for prolate ellipsoid as the magnitude of the torque $\tau_{xy}=0$. However, it is a metastable state and already a small perturbation rotates the ellipsoid back from $\alpha =90^{\circ }$ to $\alpha =0^{\circ }$. The macroscopic magnetic torque is directly proportional to the magnitude of an applied magnetic field $\vec{H_{0}}$ and volume fraction ϕ. It shows that the magnetic field opposes an applied rotation in magnetized prolate MAEs and facilitates it in the case of oblate MAEs. The Cauchy stress tensor σ alone can not account for these effects induced by an external magnetic field. The couple generated by the magnetic field $\left(\vec{M}\times \vec{H_{0}}\right)$ must be accounted for the balance of angular momentum. It dictates the symmetry conditions of the Cauchy stress tensor, as mentioned in the previous section.

## 4. Shear Deformations

We consider the amount of shear *k* as the displacement imposed on an ellipsoidal MAE sample due to homogeneous shear deformation. During the shear deformation, a sample is assumed to be confined between two parallel plates. Thus, the magneto-induced deformations are neglected. Our earlier study established that the magnetoactive elastomers with the random distribution of magnetic particles exhibit transverse isotropy along the field direction with respect to uniaxial deformations [15]. For transversely isotropic materials, the shear deformation can be distinguished between (1) Shear deformation in the plane of isotropy (the y−z plane) and (2) Shear deformation in the plane perpendicular to the plane of isotropy (the x−y plane). When shear deformation is applied in the plane of isotropy, the reinforcement does not play a role, and material yields a pure matrix response [54,55]. However, it is not true in the field-controllable materials like MAEs (see Section 4.3). The shear deformation can be further separated in the plane perpendicular to the plane of isotropy between (1) Shear deformation along the external magnetic field (in the x−y plane along *x*-direction) and (2) Shear deformation perpendicular to the external magnetic field (in the x−y plane along *y*-direction).

To describe the initial shape of an MAE sample, we use the fact that a sphere can be transformed into an ellipsoid of revolution under uniaxial deformation. Thus, the shape transformation of a unit sphere to an ellipsoid of revolution is achieved by appyling a uniaxial transformation tensor $F_{shape}$ where $\det(F_{shape})=1$ as shown in Figure 3a. The transformation tensor $F_{shape}$ is analogous to the deformation gradient tensor **F**. We use the term transformation tensor to highlight the fact that the shape transformation is a mathematical manipulation to obtain the initial shape of an MAE sample. Equivalent to the left Cauchy deformation tensor **b**, we define the tensor $b_{shape}=F_{shape}\cdot F_{shape}^{T}$.
(22)$$F_{shape}=\left[\begin{array}{ccc} \lambda_{1} & 0 & 0\\ 0 & 1/\sqrt{\lambda_{1}} & 0\\ 0 & 0 & 1/\sqrt{\lambda_{1}} \end{array}\right]$$
(23)$$b_{shape}=\left[\begin{array}{ccc} {\lambda_{1}}^{2} & 0 & 0\\ 0 & 1/\lambda_{1} & 0\\ 0 & 0 & 1/\lambda_{1} \end{array}\right]$$

Here, $\lambda_{1}$ is a stretch ratio along the *x*-axis which transforms a sphere into an ellipsoid of revolution. For $\lambda_{1}<1$ one can obtain an oblate spheroid and for $\lambda_{1}>1$ a sphere transforms into a prolate spheroid. The initial shape of an MAE sample is characterized by defining an initial aspect ratio $\gamma_{0}$ with the help of eigenvalues of $b_{shape}$. During the shear deformation, the external magnetic field is always applied along the *x*-axis (symmetry axis of an MAE sample), and the initial shape of an MAE sample is assumed to be an ellipsoid of revolution, as shown in Figure 3b–d. We choose the value of volume fraction of magnetic particles ϕ=0.3 as an optimum value [15,26,56,57], the magnetic susceptibility χ=1000, and shear modulus of 50 kPa in the absence of the applied magnetic field $\vec{H_{0}}$ following our previous work [15]. In the next Section 4.1, we apply shear deformation $F_{mech}$ to a transformed ellipsoid of revolution. Hence, the total deformation gradient tensor **F** in our formalism is given as $F=F_{mech}\cdot F_{shape}$. The corresponding total left Cauchy deformation tensor and the mechanical left Cauchy deformation tensor are calculated as $b=F\cdot F^{T}$ and $b_{mech}=F_{mech}\cdot F_{mech}^{T}$ respectively.

### 4.1. Shear Deformation along the External Magnetic Field

The shear deformation is produced by enforcing a displacement *k* along the field direction in the x−y plane, as shown in Figure 3b. The corresponding shear deformation gradient tensor is
(24)$$F_{mech}=\left[\begin{array}{ccc} 1 & k & 0\\ 0 & 1 & 0\\ 0 & 0 & 1 \end{array}\right]$$

When an MAE sample of the shape of an ellipsoid of revolution is subjected to shear deformation, it deforms into a general ellipsoid with three distinct semi-axes. Thus, to characterize its final shape, one needs two aspect ratios $\gamma_{1}$ and $\gamma_{2}$. The change in aspect ratios is calculated by determining the eigenvalues of the total left Cauchy deformation tensor **b** as $\gamma_{1}=\sqrt{\frac{e_{1}}{e_{2}}}$ and $\gamma_{2}=\sqrt{\frac{e_{1}}{e_{3}}}$. The principal invariants of the total left Cauchy deformation tensor are the coefficients of the characteristic polynomial. By solving the characteristic polynomial, the principal stretches can be expressed as the functions of invariants $\tilde{I_{1}}$ and $\tilde{I_{2}}$ (for definition see Appendix B) of the total left Cauchy deformation tensor **b** [43]. Eigenvectors of the total left Cauchy deformation tensor **b** provide angles between the principal axis of an ellipsoidal MAE and the applied magnetic field. The free energy of an ellipsoidal MAE during the shear deformation in the x−y plane can be expressed by adding Equations (Equation 2) and (Equation 20)
(25)$$\psi_{MAE}=\frac{G}{2}\left(I_{1}-3\right)-\frac{\mu_{0}\phi H_{0}^{2}}{2}\left(\frac{\cos^{2}\alpha }{R+\phi J_{a}}+\frac{\sin^{2}\alpha }{R+\phi J_{b}}\right)$$

The demagnetizing factors $J_{a}$ and $J_{b}$ along the *x* and *y*-axis are function of the aspect ratios $\gamma_{1}$ and $\gamma_{2}$. For shear deformation applied parallel to the field direction, the corresponding total stress tensor can be obtained from Equations (Equation 18) and (Equation 25)
(26)$$T_{\|}=-pI+Gb_{mech}+\zeta \left(S_{1}E_{1}+S_{2}E_{2}+S_{3}E_{3}\right)-\frac{1}{2}\tau$$
where
(27)$$\begin{array}{c} S_{1}=\phi \left(\frac{\cos^{2}\alpha }{{\left(R+\phi J_{a}\right)}^{2}}\frac{\partial J_{a}}{\partial \gamma_{1}}\;+\frac{\sin^{2}\alpha }{{\left(R+\phi J_{b}\right)}^{2}}\frac{\partial J_{b}}{\partial \gamma_{1}}\right)\\ S_{2}=\phi \left(\frac{\cos^{2}\alpha }{{\left(R+\phi J_{a}\right)}^{2}}\frac{\partial J_{a}}{\partial \gamma_{2}}\;+\frac{\sin^{2}\alpha }{{\left(R+\phi J_{b}\right)}^{2}}\frac{\partial J_{b}}{\partial \gamma_{2}}\right)\\ S_{3}=2\phi \sin\alpha \cos\alpha \left(\frac{J_{b}-J_{a}}{\left(R+\phi J_{a}\right)\left(R+\phi J_{b}\right)}\right) \end{array}$$
and
(28)$$\begin{array}{c} E_{1}=\frac{\partial \gamma_{1}}{\partial F_{mech}}\cdot F_{mech}^{T}\\ E_{2}=\frac{\partial \gamma_{2}}{\partial F_{mech}}\cdot F_{mech}^{T}\\ E_{3}=\frac{\partial \alpha }{\partial F_{mech}}\cdot F_{mech}^{T} \end{array}$$

The total stress components are
(29)$$T_{\|}=\left[\begin{array}{ccc} \sigma_{xx} & T_{xy} & 0\\ T_{xy} & \sigma_{yy} & 0\\ 0 & 0 & \sigma_{zz} \end{array}\right]$$
and the corresponding total shear stress component is
(30)$$T_{xy}=Gk+\zeta {\left(S_{1}E_{1}+S_{2}E_{2}\right)}_{xy}+\tau_{xy}\left(\frac{1}{2}-{\left(E_{3}\right)}_{xy}\right)$$

The total shear stress component $T_{xy}$ is related to the amount of shear *k* through the shear modulus *G*, parameter ζ and the initial shape of the MAE. The shear stress response $T_{xy}$ of the MAE as a function of the amount of shear *k* is shown in Figure 4a at different values of the initial aspect ratio $\gamma_{0}$ and in Figure 4b at different magnitudes of the applied magnetic field $\vec{H_{0}}$. Here, we only present results for oblate MAEs as a negligible effect is predicted for prolate MAEs during the shear deformation applied parallel to the field direction. The considerable deviation from Neo-Hookean behavior is seen for highly oblate MAEs in Figure 4a. In Figure 3b, at k=0, the symmetry axis of an oblate spheroidal MAE is parallel to the applied field direction. As mentioned in the previous section, it is a metastable state. When k>0, the applied rotation in the form of shear and the rotation imposed by the magnetic torque coincides. As a result, oblate MAEs exhibit nearly zero shear stress at small shear deformations. The total shear stress $T_{xy}$ in oblate MAEs during the shear deformation parallel to the field direction is directly proportional to the length of its symmetry axis. Consequently, for highly oblate MAEs ($\gamma_{0}<0.25$), even a negative shear stress is observed. In Figure 4a,b, the shear stress component $T_{xy}$ crosses the Neo-Hookean line at a certain critical value of $k=k_{c}$. The contribution of magnetic torque (Figure 5a) according to the symmetry condition given in Equation (Equation 16) to the Cauchy shear stress component results in a zero field-induced magnetic stress at $k=k_{c}$, as shown in Figure 5b. It is clear from Figure 4a,b that the critical value $k_{c}$ is a function of the initial aspect ratio $\gamma_{0}$ but it does not depend on the magnitude of an external magnetic field $\vec{H_{0}}$.

### 4.2. Shear Deformation Perpendicular to the External Magnetic Field

Here, we consider a shear deformation generated by forcing a displacement *k* perpendicular to the field direction, as shown in Figure 3c. Similar to the previous section, the total deformation gradient tensor and the total left Cauchy deformation tensor are $F=F_{mech}\cdot F_{shape}$, $b=F\cdot F^{T}$, respectively. The shear deformation tensor $F_{mech}$ is
(31)$$F_{mech}=\left[\begin{array}{ccc} 1 & 0 & 0\\ k & 1 & 0\\ 0 & 0 & 1 \end{array}\right]$$

As was done in Section 4.1, we use the mechanical left Cauchy deformation tensor $b_{mech}$ in the elastic part and the total left Cauchy deformation tensor **b** in the magnetic part of the total stress tensor. The total stress tensor corresponding to the shear deformation applied perpendicular to the field direction $T_{⊥}$ and corresponding shear stress component have identical form as shown in Equations (Equation 29) and (Equation 30), respectively. Here, we only present results for prolate MAEs as a negligible effect is predicted for oblate MAEs during the shear deformation applied perpendicular to the field direction. In Figure 3c, at k=0, the symmetry axis of a prolate MAE is parallel to the applied magnetic field. Unlike the previous section, it is a global equilibrium state. When shear deformation is applied, a prolate MAE generates a large amount of shear stress compared to Neo-Hookean response as the sample opposes its stretch and rotation against the direction of the applied magnetic field. The total shear stress component $T_{xy}$ increases with an increase in the initial aspect ratio $\gamma_{0}$ and with the magnitude of the applied magnetic field $\vec{H_{0}}$ as shown in Figure 4c,d, respectively. Like the previous section, in this case, the critical value $k_{c}$ exists at which the magnetic stress is zero (solid lines in Figure 5b). In Figure 5a, it is observed that the magnitude of the torque is higher during the shear deformation of oblate MAEs applied along (dashed lines) the field direction than the magnitude of the torque during the shear deformation of prolate MAEs applied perpendicular to the field direction (solid lines).

### 4.3. Shear Deformation in the Plane of Isotropy (y−z Plane)

Here, we consider a shear deformation created by forcing a displacement *k* in the plane perpendicular to the applied field direction, as shown in Figure 3d. The corresponding deformation gradient tensor is
(32)$$F_{mech}=\left[\begin{array}{ccc} 1 & 0 & 0\\ 0 & 1 & k\\ \;0 & 0 & 1 \end{array}\right]$$

The corresponding Cauchy stress tensor is
(33)$$T=\sigma =\left[\begin{array}{ccc} \sigma_{xx} & 0 & 0\\ 0 & \sigma_{yy} & \sigma_{yz}\\ 0 & \sigma_{yz} & \sigma_{zz} \end{array}\right]$$
and the Cauchy shear stress component $\sigma_{yz}$ is
(34)$$\sigma_{yz}=Gk+\frac{\zeta }{{\left(R+\phi J_{a}\right)}^{2}}{\left(\frac{\partial J_{a}}{\partial \gamma_{1}}E_{1}+\frac{\partial J_{a}}{\partial \gamma_{2}}E_{2}\right)}_{yz}$$

The shear stress response of an oblate MAEs in the y−z plane is shown in Figure 6a,b. We observe that the shear stress $\sigma_{yz}$ is not affected by the magnetic couple $\vec{M}\times \vec{H_{0}}$, which indicates that the magnetic torque does not play any role when the shear displacement is imposed in the y−z plane ($\vec{\tau }=\vec{0}$). The symmetry axis of an MAE sample (oblate and prolate) remains aligned with the magnetic field direction. Thus, there is no magnetic torque acting on an MAE sample, and the symmetry of the Cauchy stress tensor is unaffected. The shear stress deviates more pronounced from the Neo-Hooken behavior when the sample shape becomes close to spherical. The shear stress response decreases monotonically with respect to the magnitude of the applied magnetic field, as shown in Figure 6b. Similar behavior is seen for prolate MAEs. Highly oblate and prolate MAEs exhibit close to Neo-Hookean response and show a negligible effect of the magnetic field. As mentioned previously, the deviations from Neo-Hookean behavior in the isotropic (y−z) plane are observed only in the case of MAEs. These deviations are absent in the conventional transversely isotropic materials [55].

## 5. Magneto-Rheological Effect (ΔG)

The magneto-rheological (MR) effect is defined as the change of the modulus of the MAE in an external magnetic field [58]. Like in previous sections, we calculate the shear moduli of MAEs in the plane parallel to the field direction (the x−y plane) and the plane perpendicular to the field direction (the y−z plane). The shear modulus $G_{\|}$ along the field direction is calculated by taking the derivative of the shear stress component (Equation (Equation 30)) over the amount of shear *k*
(35)$$G_{\|}={\left(\frac{\partial T_{xy}}{\partial k}\right)}_{k=0}$$

Similarly, the shear modulus $G_{⊥}$ and $G_{yz}$ are calculated perpendicular to the field direction and in the y−z plane, respectively, from corresponding shear stress components. From the three different shear moduli, we obtain the relative MR-effects as
(36)$$\Delta G_{\|}=\frac{G_{\|}-G}{G}$$
(37)$$\Delta G_{⊥}=\frac{G_{⊥}-G}{G}$$
(38)$$\Delta G_{yz}=\frac{G_{yz}-G}{G}$$
where G is the zero-field shear modulus of the MAE when $\vec{H_{0}}=\vec{0}$.

Note that the shear modulus G is a function of the volume fraction of magnetic particles ϕ and elasticity of the matrix. At the same time, $G_{\|}$, $G_{⊥}$ and $G_{yz}$ are a function of the strength of the applied magnetic field $\vec{H_{0}}$, volume fraction ϕ and the initial shape $\gamma_{0}$. The % MR-effect in oblate MAEs along the field direction and prolate MAEs perpendicular to the field direction is illustrated in Figure 7a. The MR effect produced by oblate MAEs perpendicular to the field and prolate MAEs parallel to the field is negligible and not shown here. The continuous lines in Figure 7a represent the MR effect exhibited by oblate MAEs along the field direction and the dashed lines show the MR effect of prolate MAEs transverse to the field direction. Large MR effect up to 200% is observed for oblate as well prolate MAEs. Oblate MAEs yield a negative MR effect along the field direction, while prolate MAEs exhibit a positive MR effect transverse to the applied field direction. The MR effect at the initial aspect ratio $\gamma_{0}\approx 1.3$ is almost zero. As illustrated in Figure 4, at the critical value of $k=k_{c}$, MAEs exhibit Neo-Hookean behaviour. Similarly, in Figure 7a, at $\gamma_{0}\approx 1.3$, the MAE yields zero-field shear modulus. In Figure 7b, the MR effect in the plane perpendicular to the field direction is small (only a few percentages) compared to the MR effect in the x−y plane and it is negative for oblate MAEs along with prolate samples. The oblate MAEs ($\gamma_{0}<1$) show the maximum MR-effect for all the values of the applied magnetic field.

## 6. Conclusions

In this paper, we have investigated the magneto-mechanical behavior of ellipsoidal MAEs during shear deformations. In continuation of our previous work [15], additional verification of the transverse isotropy is presented by establishing the distinct direction-dependent behavior of MAEs during the shear deformations. We found that the stress-strain relationship is a strong function of the initial shape of MAEs during the applied shear deformations. Analogous to the classical transversely isotropic materials, it’s shown that the MAEs have three distinct shear moduli: $G_{\|}$ along the field direction, $G_{⊥}$ transverse to the field direction and $G_{yz}$ in the plane perpendicular to the field direction $\vec{H_{0}}$. The role of the magnetically induced torque on the symmetry of the Cauchy stress tensor has been addressed and the modified symmetry conditions have been used to construct a symmetric total stress tensor of MAEs [35,50]. The magnetic torque exerted by the applied magnetic field on ellipsoidal MAEs during shear deformation has a significant effect on the magneto-mechanical behavior of MAEs. The oblate MAEs exhibit negative MR effect $\Delta G_{\|}$ along the field direction. On the other hand, prolate MAEs yield a large positive MR-effect $\Delta G_{⊥}$ perpendicular to the field direction. These results agree qualitatively with the previous predictions where the coarse grained network model is used [59]. In recent work, the MR effect has been studied by considering the local particle rearrangement during the shear deformation, and significant MR effects have been reported [60]. In contrast, in this work, we explicitly consider the effect of the macroscopic shape of the MAE, assuming the particle distribution remains isotropic even after the deformation. Thus, we speculate that the effects emerging from the initial shape of the MAE would additionally enhance the overall MR effect. The proposed ellipsoidal approximation of MAEs can be further extended to more practical shapes like discs and cylinders. The present study of shape effect offers a basis for the synthesis of MAEs as per the commercial interests. The material model proposed in Section 2 can be implemented in the design and simulation of soft actuators on a macro-scale where ellipsoidal MAE inclusions are used for local stiffening purposes.

## Figures and Tables

**Figure 1 materials-14-03958-f001:**
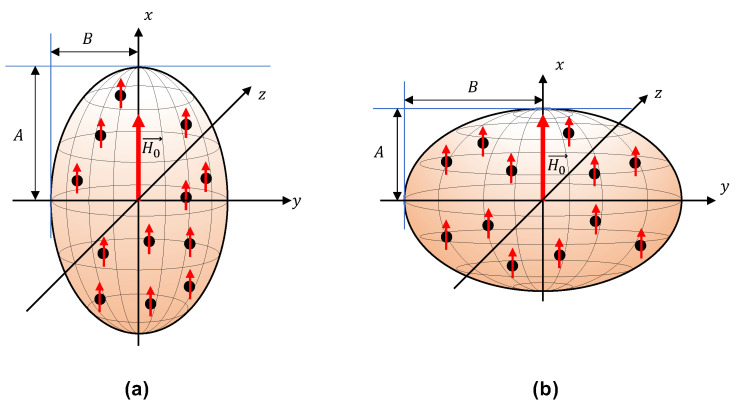
Schematics of an MAE sample with the shape of an ellipsoid of revolution with two equal semi axes B=C: (**a**) A prolate MAE sample (A>B=C). (**b**) An oblate MAE sample (A<B=C).

**Figure 2 materials-14-03958-f002:**
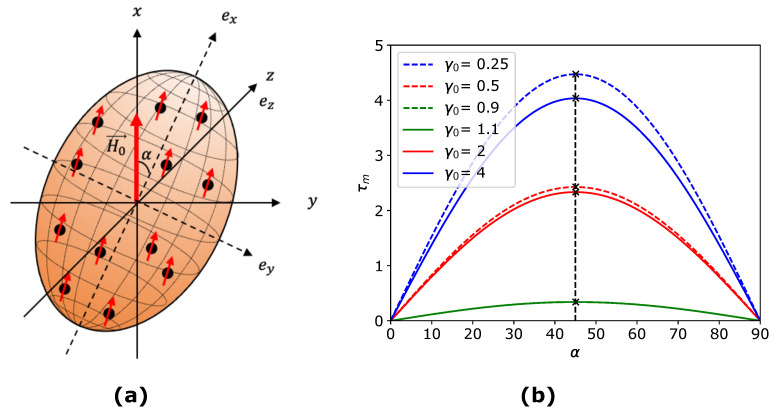
(**a**) A prolate MAE suspended freely in a uniform magnetic field $\vec{H_{0}}$ applied along the *x*-axis. (**b**) The dimensionless magnetic torque $\tau_{m}=\frac{1}{\zeta }|\tau_{xy}|$ as a function of a rotation angle α.

**Figure 3 materials-14-03958-f003:**
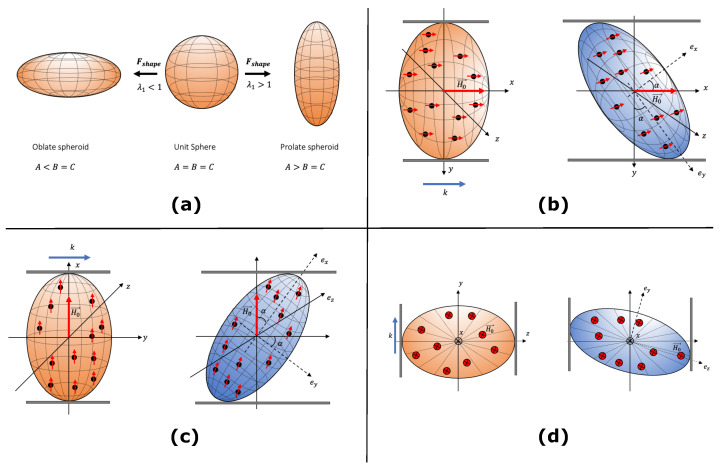
The shape transformation and shear geometries: (**a**) The transformation of a unit sphere to an ellipsoid of revolution with two equal semi-axes. (**b**) Schematics of shear deformation of an oblate MAE along the field direction. The displacement *k* is applied to the bottom plate in the *x*-direction. (**c**) Schematics of the shear deformation of a prolate MAE perpendicular to the field direction. The shear displacement *k* is applied to the top plate in the *y*-direction. (**d**) Schematics of shear deformation of an oblate MAE in the y−z plane which is perpendicular to the field direction. The circle with a cross represents the *x*-direction. The shear displacement *k* is applied to the left plate along the *y*-direction.

**Figure 4 materials-14-03958-f004:**
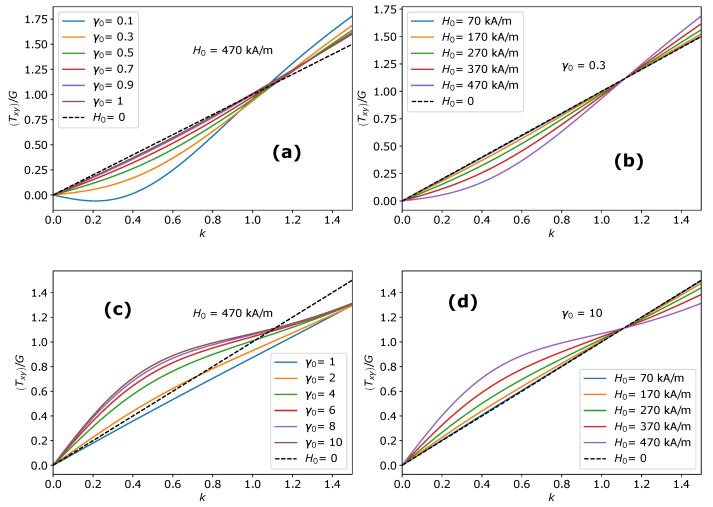
The stress-strain behavior of MAEs in the x−y plane: (**a**) Shear stress $T_{xy}$ as a function of the amount of shear *k* during the shear deformation along the field direction for oblate MAEs at different initial aspect ratios $\gamma_{0}$. (**b**) Shear stress $T_{xy}$ as a function of the amount of shear *k* during the shear deformation along the field direction for an oblate MAE ($\gamma_{0}=0.3$) at different magnitudes of the applied magnetic field $\vec{H_{0}}$. (**c**) Shear stress $T_{xy}$ as a function of the amount of shear *k* during the shear deformation perpendicular to the field direction for prolate MAEs at different initial aspect ratios $\gamma_{0}$. (**d**) Shear stress $T_{xy}$ as a function of the amount of shear *k* during the shear deformation perpendicular to the field direction for a prolate MAE ($\gamma_{0}=10$) at different magnitudes of the applied magnetic field $\vec{H_{0}}$.

**Figure 5 materials-14-03958-f005:**
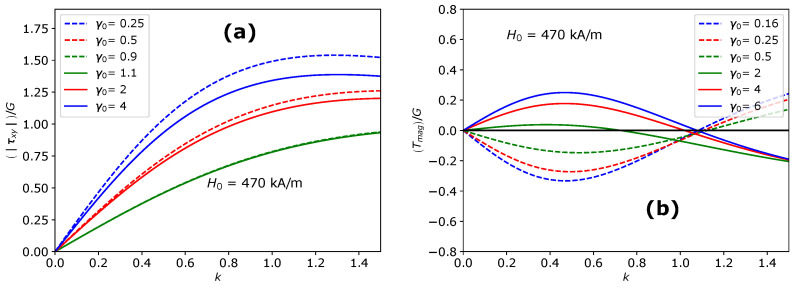
(**a**) The magnetic torque $\tau_{xy}$ as a function of the amount of shear *k* during shear deformation along the field direction (dashed lines) and perpendicular to the field direction (solid lines) for different values of the initial aspect ratio $\gamma_{0}$. (**b**) The magnetic stress $T_{mag}=\zeta {\left(S_{1}E_{1}+S_{2}E_{2}\right)}_{xy}+\tau_{xy}\left(\frac{1}{2}-{\left(E_{3}\right)}_{xy}\right)$ as a function of the amount of shear *k* during shear deformation along the field direction (dashed lines) and perpendicular to the field direction (solid lines) for different values of the initial aspect ratio $\gamma_{0}$.

**Figure 6 materials-14-03958-f006:**
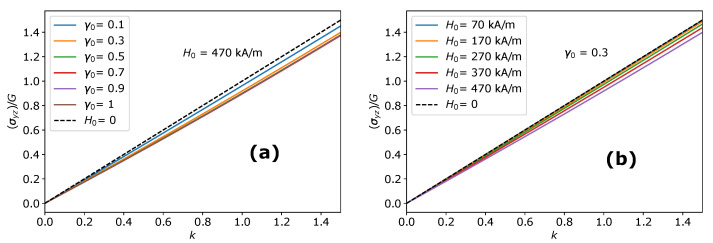
The stress-strain behavior of MAEs in the y−z plane: (**a**) Shear stress $\sigma_{yz}$ as a function of the amount of shear *k* during the shear deformation in the y−z plane along *y*-direction for oblate MAEs at different values of the initial aspect ratio $\gamma_{0}$. (**b**) Shear stress $\sigma_{yz}$ as a function of the amount of shear *k* during the shear deformation in the y−z plane along *y*-direction for an oblate MAE ($\gamma_{0}=0.3$) at different values of the magnitude of applied magnetic field $\vec{H_{0}}$.

**Figure 7 materials-14-03958-f007:**
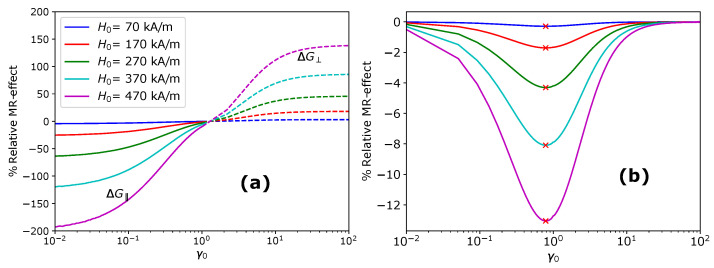
The magneto-rheological (MR) effect: (**a**) The %MR-effect in the x−y plane along the field direction (solid lines) and perpendicular to the field direction (dashed lines) as a function of the initial aspect ratio $\gamma_{0}$; (**b**) The % MR-effect in the plane of isotropy (the y−z plane) as a function of the initial aspect ratio $\gamma_{0}$.

## Data Availability

On inquiry, the data presented in this study is available from the authors.

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
