# Peer review of "Field-Induced Transversely Isotropic Shear Response of Ellipsoidal Magnetoactive Elastomers"

_materials, 2021, doi:10.3390/ma14143958_

Round 1
Reviewer 1 Report
The manuscript entitled “Field-induced transversely isotropic shear response of ellipsoidal magnetoactive elastomers” is interesting and well-written, provides new information and is publishable in Materials after a minor revision.
Suggestions:
- Make the fonts of text and of numbers within the Figures 2b, 3a-d, 4a-d, 5a-b, 6a-b and 7a-b bigger. They are difficult to read. Please, the suggestions applies also for the variables, units, the axis numbers/values etc
- The references to magnetoactive elastomers in Introduction section is too narrow. There are more relevant papers. Please include more relevant references that have worked with magneto-rheological and magneto-sensitive elastormers/rubber.
- Equation 1: The fonts of the operators tr and det should be upright, not in italics. As it is for other operators like sin, cos, tan etc
- L167: Make the division with 2 as “/2” as it is an in-line expression
- L189: Let the shear variable k be in italics as it is a variable
- L207: The fonts of the operator det should be upright, not in italics. As it is for other operators like sin, cos, tan etc
- Equation A2: The fonts of the operators tr and det should be upright, not in italics. As it is for other operators like sin, cos, tan etc
- Figure 4, 5, 6: Let the shear variable k be in italics as it is a variable
Good work!
Reviewer 2 Report
In the manuscript, there is a fundamental error in formula (13). Namely, instead of the free energy density of material as such, Authors take the magnetic energy of a finite sample. The point is that the magnetic part of the free energy density may only depend on the internal field H (or magnetic induction) or on magnetization M, but by no means on external field H0, see, for example, Landau & Lifshitz, Electrodynamics of Continuous Media, Section 31 “Thermodynamic relations in a magnetic field”. In the work under review, the MAE is implied to be filled with magnetically soft spherical microparticles. In any internal field, the induced magnetization is inevitably parallel to this field and, thus. no local torques on the particles may arise. Given that, the stress tensor remains symmetric. In terms of mechanics, rotation and/or deformation of an ellipsoidal MAE sample is due to the distribution of the surface forces – they are proportional to (Mn)2n – integrated over the surface of the ellipsoid. In this case, due to homogeneity of the internal magnetic field, there is no forces/torques in the bulk of the ellipsoid. In formal way, M (grad H) = 0, and there are no local torques since M || H.
The afore-mentioned error – it is quite fundamental – makes it meaningless to analyze the considerations which follow formula (13). The work cannot be published in the present form but might be assessed anew when and if Authors would resolve the issue.
Reviewer 3 Report
The manuscript is well-written and the theoretical calculations are well suited for this journal. However, I would like to add something here which I believe will improve the quality of this manuscript.
- It would be great if the authors add a paragraph on the various applications of magnetoactive elastomers (MAE). More importantly, how this work would enhance the MAE industry?
- Also, how would the activity change if the size of the MAE particles are in the nanometer range instead of micrometer range? I am curious to understand if this model would still work.
- How will the shape have an effect? Instead of spherical particles, if they were imagined to be spindle-shaped or rod-shaped or cylindrical-shaped particles?
Reviewer 4 Report
This paper illustrated “Field-induced transversely isotropic shear response of ellipsoidal magnetoactive elastomers”.
This paper is a major revision needed.
The comments listed below.
- This paper is all about theoretical simulations. The results are fine.
- Unfortunately, there is no discussion and analysis on the actual products of Magnetoactive elastomers (MAEs) based on this theory. As a result, it is relatively unattractive to readers.
Round 2
Reviewer 2 Report
All comments in the attached pdf file.

Reviewer 3 Report
The authors have taken my reviews seriously. The manuscript can be accepted in the present form.
Author Response
We thank the reviewer for recommending our manuscript for publication.
Reviewer 4 Report
The authors have revised the manuscript according to the comments and suggestions. This paper is ready for publication.
Author Response

(The authors gave the same response as above.)
